# Could optical coherence tomography detect monosodium urate crystal deposition at artery walls? An exploratory, phantom-based study

Javier Sánchez-Pujol[1], José Valencia[2], Juan-Miguel Ruiz-Nodar[2,3], Mariano Andrés[3,4]*

**1** Intensive Medicine unit, Elche General University Hospital, Elche, Alicante, Spain, **2** Cardiology Department, Dr. Balmis General University Hospital, Alicante Institute for Health and Biomedical Research (ISABIAL), Alicante, Spain, **3** Clinical Medicine department, Miguel Hernandez University, San Juan de Alicante, Spain, **4** Rheumatology Department, Dr. Balmis General University Hospital, Alicante Institute for Health and Biomedical Research (ISABIAL), Alicante, Spain

* mariano.andresc@umh.es

## Abstract

### Objective

Some reports suggest monosodium urate (MSU) crystals may deposit in artery walls. This exploratory study evaluated the capacity of optical coherence tomography (OCT) to detect MSU crystals in vessel phantoms.

### Methods

We used 3D-printed blood vessel phantoms made of photosensitive acrylic resins, with varying compliances. 0.1cc volume injections of centrifuged human synovial fluids containing MSU crystals, calcium pyrophosphate crystals or 0.9% saline were performed in four predefined locations in the phantoms. OCT was subsequently performed in a blinded manner by two cardiologists, experts in the technique. Two exploratory definitions for MSU crystal deposits were used, depicting a lesion with high attenuation (#1) or also with a linear morphology (#2). Diagnostic properties of both definitions were studied building 2x2 tables.

### Results

Definition #1 was found by OCT in 3/8 (37.5%) of MSU injections and in 3/16 (18.8%) of controls, what yielded a sensitivity of 37.5%, specificity of 81.3%, positive and predictive values of 50.0% and 72.2% respectively, and positive and negative likelihood ratios of 2 and 0.77. Definition #2 was only seen in 2 cases, both with MSU crystals (8.3%); this definition showed a sensitivity of 25.0%, specificity of 100%, positive and predictive values of 100.0% and 72.7%, a negative likelihood ratio of 0.8, while the positive ratio could not be calculated. Data seemed to be slightly better with Semirigid phantoms.

**Data availability statement:** The dataset that supports the present manuscript is fully available online in the Zenodo repository, within the ISABIAL community (DOI: 10.5281/zenodo.14056092).

**Funding:** The present work was funded by Alicante Institute for Health and Biomedical Research (VII Convocatoria de Ayudas para el Apoyo y Fomento de la Investigación del Instituto, ref. 2020-0321). The cited institution had no role on study design; data acquisition, analysis, or interpretation; or manuscript preparation.

**Competing interests:** Mariano Andrés declares speaking fees from Menarini (below $10,000) and an ongoing research grant from Grunenthal. The other authors declare they have no conflicts of interest in relation to this study. This does not alter our adherence to PLOS ONE policies on sharing data and materials.

## Conclusion

This preliminary, preclinical data suggest OCT could detect MSU crystals deposits in vessel walls.

Gout is the leading inflammatory arthritis in developed and developing countries, with increasing prevalence and incidence. Approximately, ~2% of adults in the UK and ~4% in the US suffer from gout. Strikingly, the disease is independently linked to developing cardiovascular events, with a derived 30% increased cardiovascular mortality [1]. How gout may foster atherosclerosis remains to be established [2], but some suggested mechanisms are hyperuricemia-related oxidate stress, endothelial dysfunction, and monosodium urate (MSU) crystal-induced inflammation. In addition, the deposition of monosodium urate (MSU) crystals in the artery wall has been proposed as an additional mechanism according to data from pathological [3] and dual-energy computed tomography (DECT) [4] studies. However, their actual presence remains uncertain. Arterial deposits were not confirmed in cadaveric gouty donors [5]; their presence was found unrelated to serum urate levels or local extent of inflammation [6], and they did not diminish under urate-lowering treatment (while musculoskeletal deposits did) [7]. Thus, the arterial deposition of MSU crystals, with derived potential repercussions [8], remains to be established.

Intracoronary optical coherence tomography (icOCT) employs a light source close to the infrared spectrum (wavelength of 1250–1350nm) with a high spatial resolution of even 10μm. The reconstructed image is based on light reflected on the tissue scanned, considering time delay and intensity measured by an interferometer. icOCT is a valid tool for achieving a direct and precise assessment of atherosclerotic vascular lesions, stent placement or intimal hyperplasia [9,10]. Moreover, recent studies suggest that icOCT can detect cholesterol crystals in the artery wall [11], as crystals often show 50–150 μm of length [12]. Accordingly, we aimed to assess whether icOCT would also be able to detect MSU crystals, with a length of about 20μm [13], in the artery wall in case they were present there.

## Methods

We designed an exploratory, preclinical study approved by the local ethical boards. Being preclinical with no human taking part in the study, no informed consent was needed. We used blood vessel phantoms made of photosensitive acrylic resins, which were 3D printed using the Stratasys J750DPA equipment (Stratasys, Rejovot, Israel), which allows a high-resolution printing of 30μm layers. The combination of resins used included BoneMatrix®, TissueMatrix®, Agilus30Clear®, VeroPureWhite®, and GelMatrix®. The latest resin was used to conform the inner space of the vessel phantoms. As depicted in the **Figure**, phantoms had two branches of different lengths. Initially, the provider supplied three different types of phantoms with varying compliances (Compliant; Slightly Compliant; Semirigid), which were tested under the OCT by two experienced interventional cardiologists using an Optis™ Mobile System equipment fitted with a Dragonfly catheter (Abbott, Abbott Park, IL). Compliant and Semirigid phantoms were found to be the most similar to human arteries and selected for the project.

Human synovial fluid samples containing MSU crystals (stored in our unit) were centrifugated, and four injections of 0.1cc volume were performed at the phantom walls, two at the short and two at the long branches of each phantom (Compliant or Semirigid), at prespecified locations as indicated in Fig 1. As controls, we used injections of calcium pyrophosphate

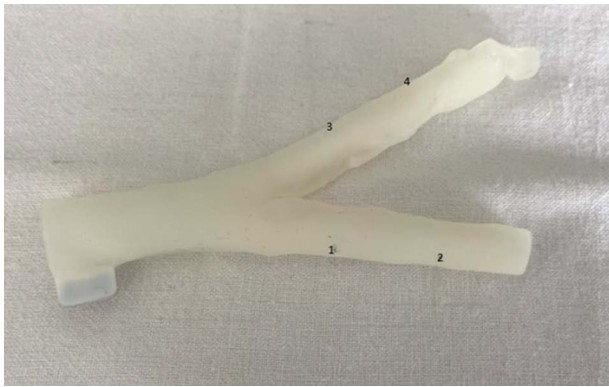

**Fig 1. An example of 3D-printed artery phantom used, with two different branches and the selected places for inoculations (numbered).**

(CPP) crystals (also from centrifugated human synovial samples) or sterile 0.9% saline. So, 24 injections (eight with MSU, eight with CPP, eight with saline) in six vessel phantoms were performed.

The injected phantoms were subsequently examined under the OCT by the cardiologists (simultaneously) (Fig 2), unaware of the inoculated sample. The primary variable was the presence of light attenuation when beams pass through the phantom layers, graded as low, medium, or high attenuation. Another variable of interest was the detection of wall lesions, registering their morphology (circular, linear, lobulated, or granular) and location.

We explored two definitions for MSU crystal deposits at OCT: first, as high attenuation images, similar to lipid deposits in atheroma plaques [10]. Second, according to cholesterol crystals in OCT [11], we required an image of linear deposition with high attenuation. We built 2x2 tables for each phantom type that allow us to calculate sensitivity, specificity, positive and negative false results rates, positive and negative predictive values, positive and negative likelihood ratios, and their 95% confidence intervals (CIs). Results were stratified regarding the phantom type (compliant or semirigid) and the control (CPP crystals or saline). Microsoft Excel 2021 (Microsoft, Albuquerque, NM) was used for calculations.

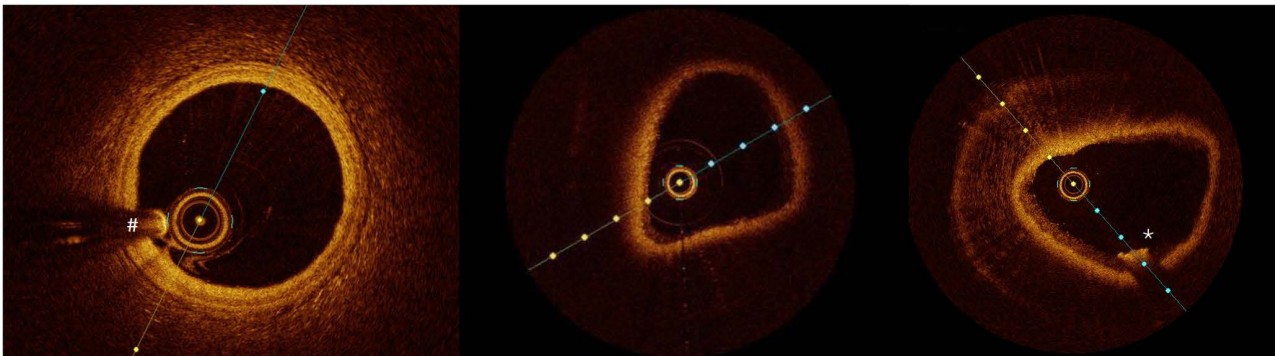

**Fig 2. OCT-rendered figures from a human coronary artery (left), a phantom with no lesions (middle), and a phantom inoculated with MSU crystals (right) showing a linear lesion with high attenuation (star).** In the left image, please note the shadow artifact (# sign) produced by the OCT catheter that should not be mistaken for a wall defect. This artifact is less evident in the phantom models.

The study has approved by our local ethics committee. As a preclinical study, no informed content was required.

## Results

Out of the 24 injections performed, the observers found wall lesions in 12 of them (50%), mostly closed to the lumen (n = 11, 91.7%). Seven lesions (58.3%) were found at the longer branch of the phantom, while five (41.7%) were at the shorter one. Six lesions (50.0%) showed a high attenuation, three (25.0%) a moderate attenuation, and three (25.0%) a low attenuation. Regarding their morphology, five lesions (41.7%) were linear, four (33.3%) were circular, two (16.7%) were granular, and one (8.3%) was lobulated.

Thus, six lesions (50.0%) fulfilled the exploratory definition #1 for possible MSU deposition (high attenuation), while two lesions (16.7%) fulfilled the definition #2 for possible MSU deposition (high attenuation of linear morphology). The results according to the crystal injected and the type of phantom used are shown in Table 1 and S1 and S2 Tables. In detail,

**Table 1. Identification of the exploratory definitions across the vessel phantoms examined, comparing between those injected with monosodium urate crystals and those acting as controls.**

| | Definition #1: *lesion with high attenuation* | | | | | | | | |
|---|---|---|---|---|---|---|---|---|---|
| | Yes, n | No, n | Total, n | Sn, % (95%CI) | Sp, % (95%CI) | PPV, % (95%CI) | NPV, % (95%CI) | PLR (95%CI) | NLR (95%CI) |
| **Whole sample** | | | | | | | | | |
| MSU crystals | 3 | 5 | 8 | 37.5 (0–77.3) | 81.3 (59.0–100.0) | 50.0 (1.7–98.3) | 72.2 (48.8–95.7) | 2.0 (0.5–7.8) | 0.8 (0.4–1.4) |
| CPP crystals or saline | 3 | 13 | 16 | | | | | | |
| Total | 6 | 18 | 24 | | | | | | |
| **Compliant phantom** | | | | | | | | | |
| MSU crystals | 1 | 3 | 4 | 25.0 (0.0–79.9) | 75.0 (38.7–100.0) | 33.3 (0.0–100.0) | 66.7 (30.3–100.0) | 1.0 (0.1–8.0) | 1.0 (0.5–2.0) |
| CPP crystals or saline | 2 | 6 | 8 | | | | | | |
| Total | 3 | 9 | 12 | | | | | | |
| **Semirigid phantom** | | | | | | | | | |
| MSU crystals | 2 | 2 | 4 | 50.0 (0.0–100.0) | 87.5 (58.3–100.0) | 66.7 (0.0–100) | 77.8 (45.1–100) | 4.0 (0.5–32.0) | 0.6 (0.2–1.6) |
| CPP crystals or saline | 1 | 7 | 8 | | | | | | |
| Total | 3 | 9 | 12 | | | | | | |
| | Definition #2: *Linear lesion with high attenuation* | | | | | | | | |
| | Yes, n | No, n | Total, n | Sn, % (95%CI) | Sp, % (95%CI) | PPV, % (95%CI) | NPV, % (95%CI) | PLR (95%CI) | NLR (95%CI) |
| **Whole sample** | | | | | | | | | |
| MSU crystals | 2 | 6 | 8 | 25.0 (0.0–61.3) | 100.0 (96.9–100.0) | 100.0 (75.0–100.0) | 72.7 (51.8–93.6) | NC | 0.8 (0.5–1.1) |
| CPP crystals or saline | 0 | 16 | 16 | | | | | | |
| Total | 2 | 22 | 24 | | | | | | |
| **Compliant phantom** | | | | | | | | | |
| MSU crystals | 0 | 4 | 4 | NC | 66.7 (39.1–86.2) | 0.0 (0.0–49.0) | 100.0 (67.6–100.0) | NC | NC |
| CPP crystals or saline | 0 | 8 | 8 | | | | | | |
| Total | 0 | 12 | 12 | | | | | | |
| **Semirigid phantom** | | | | | | | | | |
| MSU crystals | 2 | 2 | 4 | 50.0 (0.0–100.0) | 100.0 (93.8–100.0) | 100.0 (75.0–100.0) | 80.0 (50.2–100.0) | NC | 0.5 (0.2–1.3) |
| CPP crystals or saline | 0 | 8 | 8 | | | | | | |
| Total | 2 | 10 | 12 | | | | | | |

CI: confidence interval; CPP: calcium pyrophosphate; MSU: monosodium urate; NC: not calculable; NLR: negative likelihood ratio; NPV: negative predictive value; PLR: positive likelihood ratio; PPV: positive predictive value; Sn: sensitivity; Sp: specificity.

the observers detected lesion in 4 out of 8 inoculations performed with MSU crystals (50%), three in the long branch of the phantom. Three lesions were classified as with high attenuation, while the other as moderate. Two lesions depicted a circular morphology and two were linear.

Regarding the lesion with high attenuation as exploratory definition #1, this was detected in 3/8 injections with MSU crystals (37.5%) while in 3/16 control injections (18.8%), as shown in Table 1. Accordingly, the definition was specific but low sensitive, with moderate predictive values. The results appear to be better with Semirigid phantoms, regardless of the control used (S1 and S2 Tables).

When we used the exploratory definition #2 (linear image with high attenuation), this sign was only seen in 2 of the 24 injections (8.3%) performed in the phantoms; interestingly, both cases with MSU crystals (Table 1). That yielded a high specificity with good predictive values but poor sensitivity. The sign was only seen with Semirigid phantoms.

## Discussion

This is the first study evaluating the capacity of OCT to detect MSU crystal deposits in vessel phantoms and distinguish them from other elements. Our preliminary data suggest fair specificity but low sensitivity in detecting MSU, and, interestingly, this is similar to Jinnouchi and colleagues' data on cholesterol crystals by OCT [11]. The reason why MSU crystals inoculates depicted different abnormalities in the phantoms are not clear and need further assessment.

As data on how MSU crystals would be seen in the icOCT is absent, we explored two definitions, one being similar to lipid deposits (lesions with high attenuation) [10] and another also requiring a linear disposition of the lesion (as how cholesterol crystals are described in icOCT) [11]. Both signs were preliminary able to discriminate between MSU crystals inoculates and controls (PFC or saline), especially definition #2, which was only seen with MSU crystals. A possible next step might be identifying this sign in patients who underwent icOCT trying to correlate to hyperuricemia-gout background or serum urate levels. On the other hand, several MSU crystal inoculations passed unnoticed at the OCT, requiring further attention.

The results were consistent regardless of the type of control used (CPP crystals or saline). However, further studies should try other controls, such as fat extracts or basic calcium crystals, commonly present in human atheroma plaques [14]. DECT reports showed that MSU-coded deposits were closely linked to arterial calcifications [4,7]. If OCT cannot distinguish MSU deposits from lipids or calcium, the utility of icOCT will be limited for patients with gout.

We used 3D-printed arterial phantoms of acrylic resins that achieve close resemblance to human arteries at the OCT. We tested different compliances, and semirigid phantoms appeared to provide better results, which should be considered for further studies. The assessment of OCT scans blinded to substance inoculation and phantom type reinforces our findings. However, our data are preliminary and limited by the number of observations that yielded broad and overlapping confidence intervals, impeding accurate diagnostic properties estimations. Moreover, some variables could not be calculated due to null values. We inoculate crystals or saline to phantom walls to emulate deposition there. However, other mechanisms to place crystals in the phantom walls may be tested to prove the consistency of our findings. Crystals were obtained from centrifuged synovial fluids, likely accompanied by leukocytes; findings should be replicated by using pure MSU crystals (for instance, synthetics), though including leukocytes might help to get closer to a tophus-like structure [15].

We report our initial research on MSU crystal detection by OCT. Should the presence of MSU crystal in artery walls be confirmed [4], icOCT may become a valuable tool to detect crystal deposits, study the relation with the atheroma plaque, and monitor an eventual response to urate-lowering therapy.

## Supporting information

**S1 Table. Identification of the exploratory definitions across the vessel phantoms examined, comparing between those injected with monosodium urate crystals and those injected with saline acting as controls.** CI: confidence interval; CPP: calcium pyrophosphate; MSU: monosodium urate; NC: not calculable; NLR: negative likelihood ratio; NPV: negative predictive value; PLR: positive likelihood ratio; PPV: positive predictive value; Sn: sensitivity; Sp: specificity.
(DOCX)

**S2 Table. Identification of the exploratory definitions across the vessel phantoms examined, comparing between those injected with monosodium urate crystals and those injected with calcium pyrophosphate crystals acting as controls.** CI: confidence interval; CPP: calcium pyrophosphate; MSU: monosodium urate; NC: not calculable; NLR: negative likelihood ratio; NPV: negative predictive value; PLR: positive likelihood ratio; PPV: positive predictive value; Sn: sensitivity; Sp: specificity.
(DOCX)

## Acknowledgments

3D-printed vessel phantoms of acrylic resins were provided by AIJU (Technological Institute for Children's Products & Leisure, Ibi, Alicante, Spain), from where the authors would like to highlight the kind collaboration of Mr. Nacho Sandoval.

## Author contributions

**Conceptualization:** Mariano Andrés.

**Data curation:** Javier Sanchez-Pujol.

**Funding acquisition:** Mariano Andrés.

**Investigation:** Javier Sanchez-Pujol, Jose Valencia, Juan-Miguel Ruiz-Nodar, Mariano Andrés.

**Methodology:** Javier Sanchez-Pujol, Jose Valencia, Juan-Miguel Ruiz-Nodar, Mariano Andrés.

**Resources:** Jose Valencia, Juan-Miguel Ruiz-Nodar.

**Supervision:** Juan-Miguel Ruiz-Nodar, Mariano Andrés.

**Writing – original draft:** Javier Sanchez-Pujol.

**Writing – review & editing:** Javier Sanchez-Pujol, Mariano Andrés.

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
