## [Decision Letter · Decision Letter 0]

6 Nov 2024

PONE-D-23-41728Could optical coherence tomography detect monosodium urate crystal deposition at artery walls? An exploratory, phantom-based study.PLOS ONE

Dear Dr. Andrés,

Thank you for submitting your manuscript to PLOS ONE. After careful consideration, we feel that it has merit but does not fully meet PLOS ONE’s publication criteria as it currently stands. Therefore, we invite you to submit a revised version of the manuscript that addresses the points raised during the review process.

In your revised manuscript, please address the minor suggestion from Reviewer 2 regarding Figure 2. Once this is addressed your manuscript will be fit for publication.  

We look forward to receiving your revised manuscript.

Kind regards,

Emma Campbell, Ph.D

Staff Editor

PLOS ONE

on behalf of:

Concetta Zito, MD, PhD

Academic Editor

PLOS ONE<o:p></o:p>

Journal Requirements:

2. Please note that funding information should not appear in the Acknowledgments section or other areas of your manuscript. We will only publish funding information present in the Funding Statement section of the online submission form. Please remove any funding-related text from the manuscript.

“Mariano Andrés declares speaking fees from Menarini (below $10,000) and an ongoing research grant from Grunenthal. The other authors declare they have no conflicts of interest in relation to this study.”

4. In the online submission form you indicate that your data is not available for proprietary reasons and have provided a contact point for accessing this data. Please note that your current contact point is a co-author on this manuscript. According to our Data Policy, the contact point must not be an author on the manuscript and must be an institutional contact, ideally not an individual. Please revise your data statement to a non-author institutional point of contact, such as a data access or ethics committee, and send this to us via return email. Please also include contact information for the third party organization, and please include the full citation of where the data can be found.

6. Please include captions for your Supporting Information files at the end of your manuscript, and update any in-text citations to match accordingly. Please see our Supporting Information guidelines for more information: http://journals.plos.org/plosone/s/supporting-information. <o:p></o:p>

**Additional Editor Comments:**

Dear Dr Andres,

I'm glad to inform you that your manuscript "Could optical coherence tomography detect monosodium urate crystal deposition at artery walls? An exploratory, phantom-based study" has been reviewed and it has been accepted for publication in Plos One Journal.

Reviewers' comments:

Reviewer's Responses to Questions

**Comments to the Author**

1. Is the manuscript technically sound, and do the data support the conclusions?

Reviewer #1: Yes

Reviewer #2: Yes

2. Has the statistical analysis been performed appropriately and rigorously? 

Reviewer #1: Yes

Reviewer #2: Yes

3. Have the authors made all data underlying the findings in their manuscript fully available?

Reviewer #1: Yes

Reviewer #2: Yes

4. Is the manuscript presented in an intelligible fashion and written in standard English?

Reviewer #1: Yes

Reviewer #2: Yes

5. Review Comments to the Author

Reviewer #1: I appreciate the idea although sensitivity results are low. i think this preliminary work is encouraging for further studies. the manuscript data are clear. explanation of the idea and study design is good. conclusion is accurate and English language is stndard

Reviewer #2: This manuscript meets the editorial requirements and aligns well with PLOS ONE’s criteria for publication. No significant methodological concerns are noted, and there are no ethical issues raised by the study’s design or execution.

My only suggestion is related to Figure 2, which could benefit from additional clarification for readers who may not be familiar with optical coherence tomography (OCT) imaging. Specifically, the authors should emphasize that the OCT images obtained using the phantom model (middle and right panels) do not exhibit the wire artifact that is typically seen in in-vivo images (left panel). To further aid in understanding, the authors might also consider highlighting or pointing out the wire artifact in the left panel to differentiate clearly between in-vivo and phantom model imaging. This clarification would enhance the figure’s clarity, making it more accessible to a broader audience.

6. PLOS authors have the option to publish the peer review history of their article (what does this mean? ). If published, this will include your full peer review and any attached files.

**Do you want your identity to be public for this peer review?** For information about this choice, including consent withdrawal, please see our Privacy Policy .

Reviewer #1: **Yes: ** Hany Mohamed Aly

Reviewer #2: No

---

## [Author Response · Author response to Decision Letter 1]

13 Nov 2024

Response to reviewers

Editorial comments:

Response: The dataset supporting the manuscript has been made available at Zenodo online repository, as mentioned in the paper.

(Page 7, line 176): “The dataset that supports the present manuscript is fully available online in the Zenodo repository, within the ISABIAL community (DOI: 10.5281/zenodo.14056092).”

1. Thank you for stating the following in the Competing Interests section:

“Mariano Andrés declares speaking fees from Menarini (below $10,000) and an ongoing research grant from Grunenthal. The other authors declare they have no conflicts of interest in relation to this study.”

Please confirm that this does not alter your adherence to all PLOS ONE policies on sharing data and materials, by including the following statement:

"This does not alter our adherence to PLOS ONE policies on sharing data and materials.”

(as detailed online in our guide for authors http://journals.plos.org/plosone/s/competing-interests). If there are restrictions on sharing of data and/or materials, please state these. Please note that we cannot proceed with consideration of your article until this information has been declared.

Response: The indicated statement has been added to the Disclosure section.

Response: As indicated, we have removed the Ethics section that was located at the end of the manuscript.

Reviewer #1: I appreciate the idea although sensitivity results are low. i think this preliminary work is encouraging for further studies. the manuscript data are clear. explanation of the idea and study design is good. conclusion is accurate and English language is stndard

Response: We thank the reviewer’s opinion regarding our work.

Reviewer #2: This manuscript meets the editorial requirements and aligns well with PLOS ONE’s criteria for publication. No significant methodological concerns are noted, and there are no ethical issues raised by the study’s design or execution.

My only suggestion is related to Figure 2, which could benefit from additional clarification for readers who may not be familiar with optical coherence tomography (OCT) imaging. Specifically, the authors should emphasize that the OCT images obtained using the phantom model (middle and right panels) do not exhibit the wire artifact that is typically seen in in-vivo images (left panel). To further aid in understanding, the authors might also consider highlighting or pointing out the wire artifact in the left panel to differentiate clearly between in-vivo and phantom model imaging. This clarification would enhance the figure’s clarity, making it more accessible to a broader audience.

Response: Thanks for the thoughtful suggestion, which we fully agree with. Accordingly, we have introduced a clarification in the Figure 2 legend.

(Figure legends): “Figure 2. OCT-rendered figures from a human coronary artery (left), a phantom with no lesions (middle), and a phantom inoculated with MSU crystals (right) showing a linear lesion with high attenuation (star). In the left image, please note the shadow artifact (# sign) produced by the OCT catheter that should not be mistaken for a wall defect. This artifact is less evident in the phantom models.”

---

## [Decision Letter · Decision Letter 1]

25 Feb 2025

Could optical coherence tomography detect monosodium urate crystal deposition at artery walls? An exploratory, phantom-based study.

PONE-D-23-41728R1

Dear Dr. Andrés,

We’re pleased to inform you that your manuscript has been judged scientifically suitable for publication and will be formally accepted for publication once it meets all outstanding technical requirements.

Kind regards,

Emmanuel Kokori, M.D

Academic Editor

PLOS ONE

Reviewers' comments:

Reviewer's Responses to Questions

**Comments to the Author**

1. If the authors have adequately addressed your comments raised in a previous round of review and you feel that this manuscript is now acceptable for publication, you may indicate that here to bypass the “Comments to the Author” section, enter your conflict of interest statement in the “Confidential to Editor” section, and submit your "Accept" recommendation.

Reviewer #2: All comments have been addressed

2. Is the manuscript technically sound, and do the data support the conclusions?

Reviewer #2: Yes

3. Has the statistical analysis been performed appropriately and rigorously? 

Reviewer #2: Yes

4. Have the authors made all data underlying the findings in their manuscript fully available?

Reviewer #2: Yes

5. Is the manuscript presented in an intelligible fashion and written in standard English?

Reviewer #2: Yes

6. Review Comments to the Author

Reviewer #2: Thank you for alborating on the images. All comments have been addressed

Thank you for alborating on the images. All comments have been addressed

7. PLOS authors have the option to publish the peer review history of their article (what does this mean? ). If published, this will include your full peer review and any attached files.

**Do you want your identity to be public for this peer review?** For information about this choice, including consent withdrawal, please see our Privacy Policy .

Reviewer #2: **Yes: ** Giuseppe Andò, MD PhD

---

## [Editor Report · Acceptance letter]

PONE-D-23-41728R1

PLOS ONE

Dear Dr. Andrés,

I'm pleased to inform you that your manuscript has been deemed suitable for publication in PLOS ONE. Congratulations! Your manuscript is now being handed over to our production team.

Kind regards,

on behalf of

Dr. Emmanuel Kokori

Academic Editor

PLOS ONE